# Are all layers created equal:
# A neural collapse perspective

Jinxin Zhou, Jiachen Jiang, Zhihui Zhu
Ohio State University,
`zhou.3820@osu.edu`, `jiang.2880@osu.edu`, `zhu.3440@osu.edu`

Understanding how features evolve layer by layer is crucial for uncovering the inner workings of deep neural networks. *Progressive neural collapse*, where successive layers increasingly compress within-class features and enhance class separation, has been primarily studied empirically in small architectures on simple tasks or theoretically within linear network contexts. However, its behavior in larger architectures and complex datasets remains underexplored. In this work, we extend the study of progressive neural collapse to larger models and more complex datasets, including clean and noisy data settings, offering a comprehensive understanding of its role in generalization and robustness. Our findings reveal three key insights: 1. Layer inequality: Deeper layers significantly enhance neural collapse and play a vital role in generalization but are also more susceptible to memorization. 2. Depth-dependent behavior: In deeper models, middle layers contribute minimally due to a diminished neural collapse enhancement leading to redundancy and limited generalization improvements, which validates the effectiveness of layer pruning. 3. Architectural differences: Transformer models outperform convolutional models in enhancing neural collapse on larger datasets and exhibit greater robustness to memorization, with deeper Transformers reducing memorization while deeper convolutional models show the opposite trend. These findings provide new insights into the hierarchical roles of layers and their interplay with architectural design, shedding light on how deep neural networks process data and generalize across challenging conditions.

## 1. Introduction

Deep learning has become the de facto choice for a wide range of machine learning applications, including image recognition [1, 2], language modeling [3–5], and scientific computing [6, 7]. A key factor driving its widespread applications in diverse real-world scenarios is its superior generalization and robustness when handling massive datasets, even in the presence of human annotation errors or machine-induced corrupted inputs. However, despite their widespread success, the underlying reasons for the remarkable generalization and robustness abilities of deep networks remain poorly understood. Much of their success has been attributed to the ability to learn hierarchical representations, which enables deep learning models to capture complex patterns across different layers [8]. Yet, precisely characterizing hierarchical representations in high-dimensional space remains a significant challenge, leaving the mechanisms behind their generalization and robustness only partially explained. In this paper we are motivated by the following question:

*how to characterize hierarchical representations, and how do they differ in the presence of noisy data?*

Papyan et al. [9] empirically identified an intriguing phenomenon termed *Neural Collapse* ($\mathcal{NC}$) for the balanced multi-class classification tasks. During the terminal phase of training, once the training error reaches zero, both the last-layer features and the final linear classifier converge to a highly symmetric and structured geometric configuration. Specifically, the last-layer features collapse to their corresponding class means ($\mathcal{NC}_1$), and the class-mean features themselves are maximally distant, forming a simplex equiangular tight frame (ETF) structure ($\mathcal{NC}_2$). Simultaneously, the classifier

weights align perfectly with the centered class-mean features, up to a scaling factor ($\mathcal{NC}_3$). Consequently, this geometric structure leads the classifier to make predictions by selecting the class with the nearest train class mean ($\mathcal{NC}_4$).

Neural Collapse offers a mathematically elegant characterization of the learned representations in the penultimate layer of deep classification models, independent of network architecture and dataset. While the research in this field has enhanced the understanding of how deep neural networks functions from different perspectives, previous studies mainly focus on last-layer features or examines intermediate features using relatively small network architectures, such as MLP, VGG, and shallow ResNet, and on simpler datasets like MNIST, Fashion-MNIST, and CIFAR-10. For example, He & Su [10] suggests that intermediate layers in deep networks decrease the within-class variability at a constant geometric rate, suggesting that "all layers are created equal" in reducing separation fuzziness. However, this phenomenon has primarily been observed in simple architectures and datasets, raising questions about its applicability to more complex, real-world scenarios. In particular, it has been very recently observed in large models that middle layers are more redundant and can be removed without significant performance degradation [11, 12], which indicates that such models may not obey the equal-separation rule across layers. This gap raises an important question: **Are all layers created equal across different architectures and datasets?**

Moreover, since real-world datasets often contain noise from annotation errors and sensor imperfections, this noise can propagate through the layers, distorting feature hierarchies and impairing the network's ability to generalize. Understanding how noise affects the evolution of features across layers is essential for improving the reliability of neural networks, such as determining the optimal depth and selecting more robust models. However, most existing theoretical and empirical studies on neural collapse either rely on the unconstrained feature model assumption, treating all data samples equally, or pay limited attention to the process by which neural networks transform noisy data points into a simplex ETF structure during forward propagation. This raises questions on **how the presence of noise affects the behavior of each layer for feature compression and separation?**

**Contributions.** In this work, we conduct an extensive empirical investigation across diverse vision datasets to address these questions, focusing on the intermediate representations of both convolutional and Transformer models from neural collapse perspectives. Our contributions are as follows:

1. **Unequal layer contributions:** Deeper layers excel at enhancing neural collapse, playing a critical role in generalization but are also more prone to memorization, regardless of architecture, model depth, or dataset complexity. This finding contrasts with the results reported in [10], which suggested that all layers contribute equally to data separation. Moreover, the positive correlation between neural collapse enhancement and generalization performance, along with its negative relationship with robustness, offers valuable insights into the distinct roles of individual layers.

2. **Impact of model depth:** In deeper models, middle layers exhibit weaker neural collapse enhancement with an emergent plateau region. These layers contribute minimally, often making deeper models redundant, which explains the limited generalization gains observed with increasing depth and the success of layer pruning technique in large language models.

3. **Cross-architecture comparison:** Our analysis reveals that Transformer models outperform convolutional models in enhancing neural collapse on larger datasets and generalization and are less prone to memorization. Notably, deeper Transformer models exhibit reduced memorization, whereas deeper convolutional models show an increasing trend, highlighting a fundamental difference in their learning mechanisms.

## 2. Related Works

**Last-layer neural collapse.** The $\mathcal{NC}$ phenomenon was first discovered in [9].Under the assumption of the *unconstrained feature model*(UFM) [13, 14], which treats last-layer features as free optimization variables, a series of theoretical studies have validated the existence of the $\mathcal{NC}$ phenomenon. For example, studies such as [13–23] demonstrated that the global minimizers satisfy the $\mathcal{NC}$ proper-

ties for a family of loss functions, including cross-entropy loss, mean-square-error loss, and label-smoothing loss, among others, when the last-layer feature dimension is not smaller than the number of classes. Moreover, when the number of classes is sufficiently large, the studies [24, 25] proved that the last-layer features satisfy generalized $\mathcal{NC}$ properties. Beyond UFM, Tirer et al. [20] and Dang et al. [26] characterize the global optimality of a two-layers models and multi linear layer models, respectively. Sukenik [27] extended UFM to arbitrary non-linear layers and proved that $\mathcal{NC}$ emerges after a certain layer for binary classification. These work not only contribute to a new understanding of the working of DNNs but also has also inspired the development of novel techniques across various applications, such as imbalanced learning [28, 29], transfer learning [28, 30–32], and adversarial robustness [33].

**Intermediate neural collapse.** While $\mathcal{NC}$ was initially introduced to describe the configurations of last-layer features, recent studies have extended its investigation to intermediate representations. Tirer et al. [34] provided a theoretical analysis showing that the within-class variability ($\mathcal{NC}_1$) metric decreases monotonically along the gradient flow across layers when the network is trained using cascade learning, where a new layer is added on top of the pre-trained network at each step. However, this theoretical result does not fully align with the more common practice of training models in an end-to-end manner. Apart from the theoretical results, some empirical studies [10, 35–37] suggest that the within-class variability ($\mathcal{NC}_1$) of intermediate features decreases monotonically as layers progress deeper into the network. Similarly, research by [38, 39] demonstrates that intermediate layers gradually improve the nearest class-center accuracy ($\mathcal{NC}_4$). Rather than focusing on individual $\mathcal{NC}$ properties, recent works [36, 40, 41] have extended the analysis to encompass all $\mathcal{NC}$ properties across intermediate layers. However, all of these studies investigate intermediate $\mathcal{NC}$ using relatively small network architectures, such as MLP, VGG, and shallow ResNet, and simpler datasets like MNIST, Fashion-MNIST, and CIFAR-10, which limits the generalizability of their findings to more complex architectures and datasets.

# 3. The Problem Setup

**Notations and Organization.** Throughout the paper, we use bold lowercase and upper letters, such as $\boldsymbol{a}$ and $\boldsymbol{A}$, to denote vectors and matrices, respectively. Not-bold letters are reserved for scalars. The symbols $\boldsymbol{I}_K$ and $\boldsymbol{1}_K$ respectively represent the identity matrix and the all-ones vector with an appropriate size of $K$, where $K$ is some positive integer. We use $[K] := \{1; 2; \cdots; K\}$ to denote the set of all indices up to $K$. For any matrix $\boldsymbol{A} \in \mathbb{R}^{n_1 \times n_2}$, we write $A = [\boldsymbol{a}_1 \quad \cdots \quad \boldsymbol{a}_{n_2}]$, so that $\boldsymbol{a}_i$ ($i \in [n_2]$) denotes the $i$-th column vector of $\boldsymbol{A}$.

## 3.1. Basics of Deep Neural Networks

Consider a multi-class classification problem with $K$ classes, where each class has $n$ samples $\{\boldsymbol{x}_{k,i}, \boldsymbol{y}_t\}$ i.i.d. sampled from some unknown distributions $\mathcal{P}$. The label of the $i$-th sample $\boldsymbol{x}_{k,i} \in \mathbb{R}^D$ in the $k$-th class is represented by a one-hot vector $\boldsymbol{y}_k \in \mathbb{R}^K$ with unity only in $k$-th entry ($1 \leq k \leq K$). To learn the underlying mapping from the input instance $\boldsymbol{x}_{k,i}$ to their corresponding label $\boldsymbol{y}_k$, deep neural networks stand out among a family of parameterized functions due to their outstanding performance. A typical deep neural network $\Phi_{\boldsymbol{\Theta}}(\boldsymbol{x}_{k,i})$ comprises a encoder network $\phi_{\theta_L}(\boldsymbol{x}_{k,i})$ with $L$ non-linear layers arranged in a layer-wise fashion, followed by a linear classifier $\{\boldsymbol{W}_{L+1}, \boldsymbol{h}_{L+1}\}$, which can be expressed as:

$$\Phi_{\boldsymbol{\Theta}}(\boldsymbol{x}_{k,i}) = \boldsymbol{W}_{L+1} \cdot \phi_{\theta_L}(\boldsymbol{x}_{k,i}) + \boldsymbol{b}_{L+1}; \tag{1}$$

$$\text{and} \quad \phi_{\theta_l}(\boldsymbol{x}_{k,i}) = \sigma\left(\boldsymbol{W}_l \cdot \phi_{\theta_{l-1}}(\boldsymbol{x}_{k,i}) + \boldsymbol{b}_l\right), \quad \text{where} \quad 1 \leq l \leq L; \tag{2}$$

$$\text{and} \quad \phi_{\theta_0}(\boldsymbol{x}_{k,i}) = \boldsymbol{x}_{k,i}, \tag{3}$$

where $\boldsymbol{W}_{L+1}$ and $\boldsymbol{b}_{L+1}$ represents the weight and bias terms of last-layer linear classifier, respectively. For a $L$-layer encoder network $\phi_{\theta_L}(\boldsymbol{x}_{k,i})$, each layer (e.g., the $l$-th layer where $1 \leq l \leq L$) is composed of an affine transformation $\{\boldsymbol{W}_l, \boldsymbol{b}_l\}$, followed by a nonlinear activation $\sigma(\cdot)$ and some normalization functions (e.g., BatchNorm), to extract hierarchical expressive features $\{\phi_{\theta_l}(\boldsymbol{x}_{k,i})\}_{l=1}^L$

from the underlying input instance $\boldsymbol{x}_{k,i}$. For simplicity, we use $\boldsymbol{\Theta}$ to denote all parameters $\{\boldsymbol{W}_l, \boldsymbol{b}_l\}_{l=1}^{L+1}$ of the entire networks and $\theta_l$ to denote the entire parameters of the first $l$-th layers in the encoder networks for $\forall l \in [L]$, where $\theta_L$ represents the all parameters $\{\boldsymbol{W}_l, \boldsymbol{b}_l\}_{l=1}^{L}$ of the encoder networks. To learn an effective deep classifier, the network parameters, the network parameters $\boldsymbol{\Theta}$ are optimized by minimizing the following empirical risk over the entire $N = nK$ training samples:

$$\boldsymbol{\Theta} := \{\boldsymbol{W}_l, \boldsymbol{b}_l\}_{l=1}^{L+1} := \{\theta_L, \boldsymbol{W}_{L+1}, \boldsymbol{b}_{L+1}\} := \arg\min_{\boldsymbol{\Theta}} \frac{1}{nK} \sum_{t=1}^{K} \sum_{i=1}^{n} \mathcal{L}\left(\boldsymbol{\Phi}_{\boldsymbol{\Theta}}(\boldsymbol{x}_{k,i}), \boldsymbol{y}_k\right),$$

where $\mathcal{L}(\boldsymbol{\Phi}_{\boldsymbol{\Theta}}(\boldsymbol{x}_{k,i}), \boldsymbol{y}) : \mathbb{R}^K \times \mathbb{R}^K \to \mathbb{R}^+$ is a specified loss function which appropriately measure the discrepancy between the prediction $\boldsymbol{\Phi}_{\boldsymbol{\Theta}}(\boldsymbol{x}_{k,i})$ and its corresponding label $\boldsymbol{y}_k$.

### 3.2. Neural Collapse

Neural Collapse ($\mathcal{NC}$) is an universal phenomenon observed in the last-layer features and the linear classifier of deep neural networks trained on classification problems. During the terminal phase of training (TPT), when the training reaches perfect accuracy, several appealing properties emerge, including the within-class variability collapse and the maximal equiangular separation among class centers of features from different classes. For notation simplification, we simplify the notation of $i$-th layer features $\phi_{\theta_l}(\boldsymbol{x}_{k,i})$ as $\boldsymbol{h}_{l,k,i}$ for $\forall l \in [L]$, $k \in [K]$ and $i \in [n]$. Additionally, we denote the global mean $\overline{\overline{\boldsymbol{h}}}_l$ and $k$-th class mean $\overline{\boldsymbol{h}}_{l,k}$ of $i$-th layer features as $\overline{\overline{\boldsymbol{h}}}_l = \frac{1}{nK} \sum_{k=1}^{K} \sum_{i=1}^{n} \boldsymbol{h}_{l,k,i}$ and $\overline{\boldsymbol{h}}_{l,k} = \frac{1}{n} \sum_{i=1}^{n} \boldsymbol{h}_{l,k,i}$. Therefore, these two $\mathcal{NC}$ properties[1] of intermediate layers (e.g. $l \in [L]$) features can be expressed as follows:

- **Within-class compression** ($\mathcal{NC}_1$). In each class, the last-layer features converges to their corresponding class-mean centers with zero variability, $\sigma_{L,k} = \frac{1}{n} \sum_{i=1}^{n} \left\| \boldsymbol{h}_{L,k,i} - \overline{\boldsymbol{h}}_{L,k} \right\|_2^2 \to 0, \quad \forall k \in [K], \ i \in [n]$. Inspired by foundational works [42, 43], $\mathcal{NC}_1$ was originally quantified for the last-layer features using an inverse *signal-to-noise ratio* (SNR), which depends on the ratio of within-class variability to between-class variability. To measure the within-class compression of intermediate features, we employ the class-distance normalized variance (CDNV) proposed by [32] and extend it to the intermediate features:

$$\mathcal{NC}_1 = \sum_{k=1}^{K} \sum_{k' \neq k}^{K} \text{CDNV}_{l,k,k'} := \sum_{k=1}^{K} \sum_{k' \neq k}^{K} \frac{\sigma_{l,k}^2 + \sigma_{l,k'}^2}{2 \left\| \overline{\boldsymbol{h}}_{l,k} - \overline{\boldsymbol{h}}_{l,k'} \right\|_2^2}, \quad \forall k \neq k', l \in [L]. \tag{4}$$

The intermediate feature compression is then characterized by the minimization of this quantity: $\sum_{k=1}^{K} \sum_{k' \neq k}^{K} \text{CDNV}_{l,k,k'} \to 0, \forall l \in [L]$. This alternative measurement is faithful[2] to the one used in the [10] but it is typically more numerically stable and computationally efficient. In contrast, the metric in the [10] requires computing the pseudoinverse of the between-class covariance matrix, which can be computationally demanding.

- **Maximal between-class separation** ($\mathcal{NC}_2$). At the last layer, the class-mean centers $\overline{\boldsymbol{h}}_{L,k}$ centered at the global-mean center $\overline{\overline{\boldsymbol{h}}}_L$ are maximally and equally distant, which exhibits an elegant Simplex Equiangular Tight Frame (ETF) structure. Therefore, for the intermediate layers, given $\overline{\boldsymbol{H}}_L = \left[ \overline{\boldsymbol{h}}_{L,1} - \overline{\overline{\boldsymbol{h}}}_L \quad \cdots \quad \overline{\boldsymbol{h}}_{L,K} - \overline{\overline{\boldsymbol{h}}}_L \right]$, we define the intermediate between-class separation as

$$\mathcal{NC}_2 = \left\| \frac{\overline{\boldsymbol{H}}_L^T \overline{\boldsymbol{H}}_L}{\left\| \overline{\boldsymbol{H}}_L^T \overline{\boldsymbol{H}}_L \right\|_F} - \frac{1}{\sqrt{K-1}} \left( \boldsymbol{I}_K - \frac{1}{K} \mathbf{1}_K \mathbf{1}_K^T \right) \right\|_F \to 0. \tag{5}$$

---

[1] In this work, we focus exclusively on the geometry of features, whereas $\mathcal{NC}_3$ and $\mathcal{NC}_4$ pertain to the alignment between features and the final classifier, which is not relevant to the study of intermediate neural collapse. Additionally, the linear probing accuracy implicitly captures the remaining two properties.

[2] Although our $\mathcal{NC}_1$ metric differs slightly from that used in [10], we observe a similar geometric rate of improvement in intermediate neural collapse when using limited-capacity models on small datasets, such as ResNet18 on CIFAR-10, as shown in Figure 1(a).

# 4. Results

In this section, we address the following three questions from the perspective of neural collapse:

1. Are layers at different depths created equal?

2. Are layers in models of varying depths created equal?

3. Are layers from different architectures created equal?

## 4.1. Are All Layers Created Equal under Clean Data?

To explore these questions, we analyze intermediate neural collapse using convolutional networks (e.g., ResNet) and transformer networks (e.g., Swin-Transformer) across various datasets under clean data (Section 4.1) and noisy data conditions (Section 4.2), examining its correlation with generalization and robustness. Additional results and details are provided in the Appendix.

**1. No, deeper layers excel at enhancing neural collapse.** As illustrated in Figure 1 and Figure 2, intermediate neural collapse consistently improves as the model progresses through deeper layers, although the rate of improvement decreases with increasing dataset complexity, from CIFAR-10 to CIFAR-100 to ImageNet. Moreover, our results show that deeper layers enhance neural collapse more rapidly than shallow layers, as evidenced by the slope of each layer. This finding contrasts with the geometric rate decay of within-class compression across layers reported in [10], which was primarily observed in models with limited capacity on small datasets. The stronger neural collapse enhancement in the final layers supports the widely held belief that shallower layers primarily focus on learning universal and transferable features, while deeper layers specialize in capturing task-specific features. Recall that both within-class compression and between-class separation are evaluated based on features grouped according to the label assignments. To explore its relationship with generalization performance, we performed linear probing on the intermediate blocks of various models. Our findings reveal that stronger neural collapse enhancement leads to greater generalization improvements, as demonstrated on the ImageNet dataset in Figure 1(i) and Figure 2(i). While the performance improvement in the final layers of ResNet models is relatively small on CIFAR datasets, we attribute this to the simplicity of CIFAR tasks for ResNet models, where well-structured features are learned earlier in the network, leaving limited scope for further improvement.

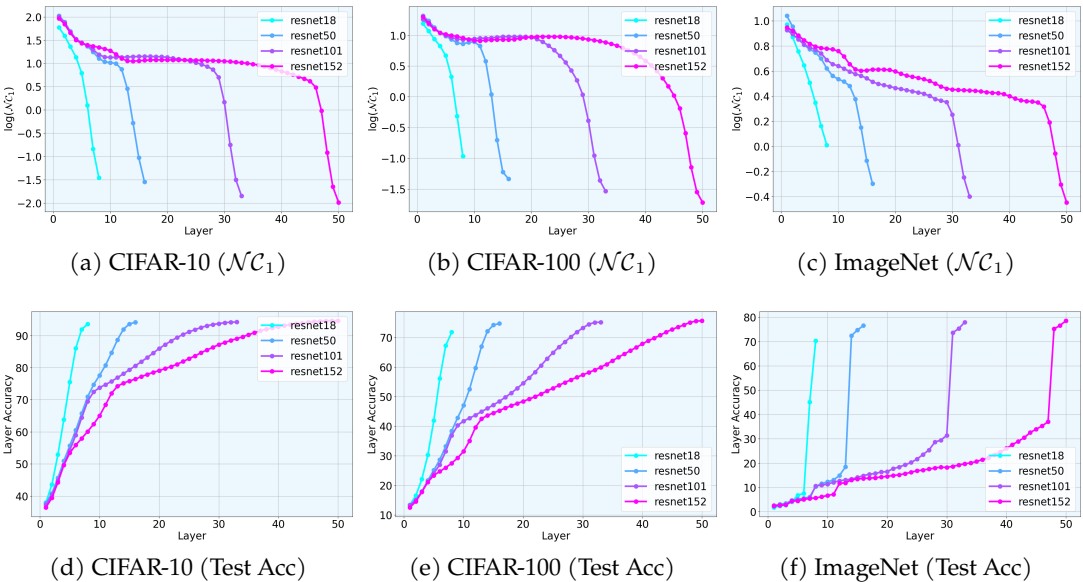

(a) CIFAR-10 ($\mathcal{NC}_1$)  (b) CIFAR-100 ($\mathcal{NC}_1$)  (c) ImageNet ($\mathcal{NC}_1$)

(d) CIFAR-10 (Test Acc)  (e) CIFAR-100 (Test Acc)  (f) ImageNet (Test Acc)

Figure 1: **The $\mathcal{NC}_1$ and linear-probing accuracy of ResNet models on various datasets.**

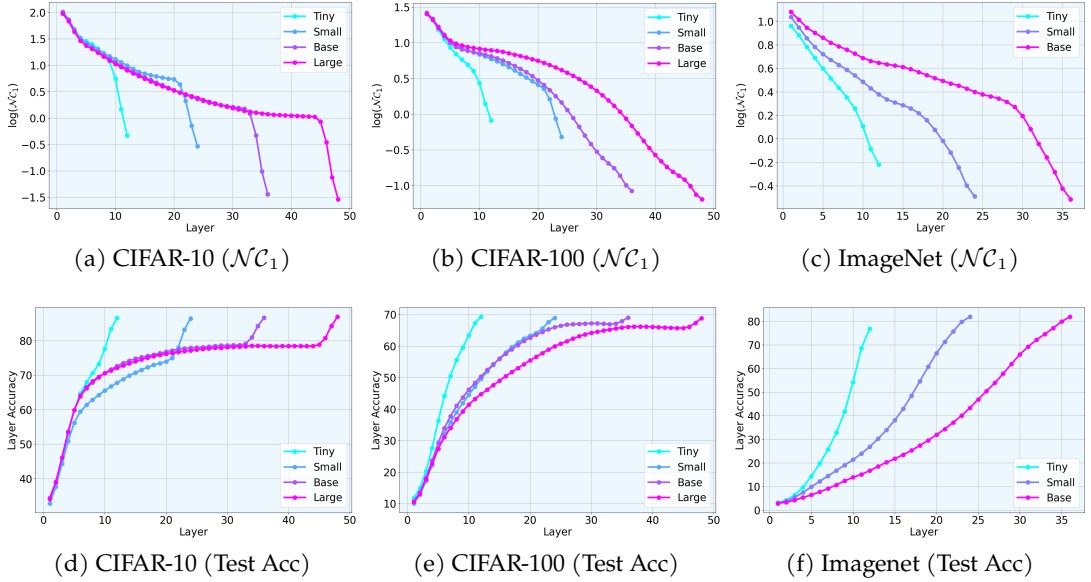

Figure 2: **The $\mathcal{NC}_1$ and linear-probing accuracy of Swin-Transformers on different datasets.**

**2. No, middle layer in a deeper model shows diminished neural collapse improvement with an emergent plateau region.** As shown in Figure 1 and Figure 2, within a dataset, increasing model depth slows down the rate of neural collapse improvement for individual layers. Moreover, once the model's capacity sufficiently matches the dataset complexity, an interesting pattern emerges as depicted in Figure 1(a,b). After an initial improvement in within-class compression in the early layers, a plateau occurs in the middle layers. Beyond this plateau, the final layers exhibit a renewed and continuous improvement in within-class compression. Continuously increasing the depth of an already sufficiently deep model introduces more middle layers stuck on the plateau. Furthermore, Figure 7 in the appendix shows that models within the same architecture exhibit a consistent relationship between intermediate neural collapse and linear probing accuracy. Notably, increasing model depth introduces additional redundant middle layers and results in only marginal improvements in final-layer generalization once the plateau phase is reached. For example, ResNet50, ResNet101, and ResNet152 all reach the plateau phase on the CIFAR-10 and CIFAR-100 datasets, whereas ResNet18 does not. The accuracies of ResNet50, ResNet101, and ResNet152 on the CIFAR-10 dataset are $95.55\%$, $95.58\%$, and $95.58\%$, respectively, while on the CIFAR-100 dataset, they achieve $75.91\%$, $75.93\%$, and $76.01\%$. In contrast, ResNet18 achieves lower accuracies of $94.72\%$ on CIFAR-10 and $74.52\%$ on CIFAR-100. Although deeper models outperform ResNet18, the differences among the deeper models themselves are negligible. This observed plateau and its relationship with generalization suggest that increasing the depth of an already sufficiently deep model primarily introduces redundant layers that focus on learning invariant, task-irrelevant, universal features, contributing little to overall performance gains, which offers a novel explanation for the success of recent works [11, 12] that prune middle layers in large models without significant performance degradation.

**3. No, the neural collapse enhancement of layers from different architectures varies across datasets.** As shown in Figure 1 and Figure 2, ResNet layers demonstrate superior within-class compression and between-class separation compared to Swin-Transformer layers on smaller datasets such as CIFAR-10 and CIFAR-100, despite Swin-Transformer layers having twice as many parameters per block. Consequently, the ResNet architecture reaches the plateau phase with fewer layers than the Swin-Transformer. In contrast, on larger datasets such as ImageNet, Swin-Transformer layers outperform ResNet layers in both within-class compression and between-class separation. To the best of our knowledge, this dynamic compression and separation abilities of different architectures across various datasets has not been previously explored. By analyzing linear probing accuracy, we find that, for models of comparable depth, the accuracy of middle layers coincides with intermediate neural collapse degree. This suggests that the intermediate neural collapse serves as

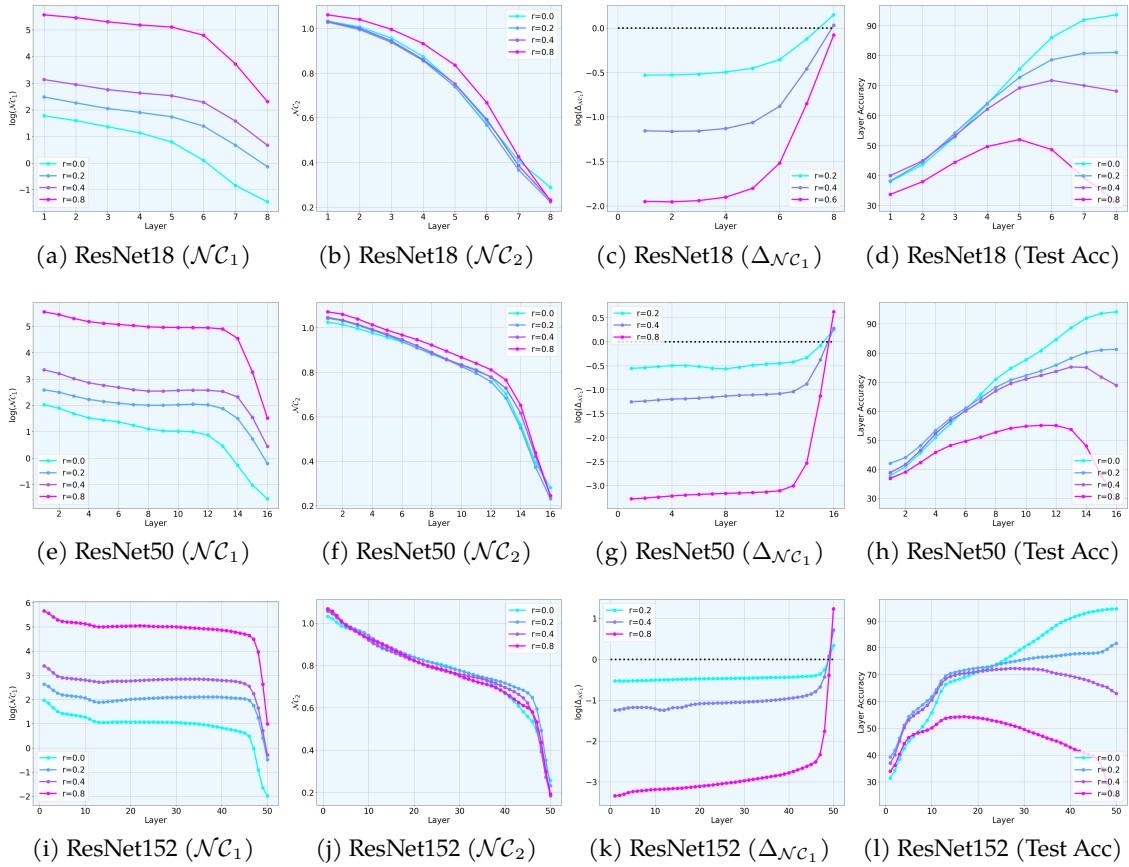

Figure 3: **The intermediate $\mathcal{NC}$, memorization ratio and linear-probing accuracy of ResNet models on random labeled CIFAR-10 dataset.** Here, $r$ denotes the percentage of randomly labeled data.

a valuable indicator of layer effectiveness. The stronger intermediate compression and separation observed in ResNet models on CIFAR-10 and CIFAR-100, and in Swin-Transformer models on ImageNet, suggest that these architectures are more efficient at extracting and organizing meaningful features within shallow or moderately deep configurations. This provides a novel explanation for the superior performance of ResNet models on small-sized, low-resolution datasets [44–46], and of Swin-Transformer models on larger datasets, which offers valuable insights for future model design.

## 4.2. Are All Layers Created Equal under Noisy Data?

Since the work [47], which demonstrate that DNNs can memorize random labels, the performance of DNNs on noisy labels has been leveraged to understand their generalization and memorization properties [48–51]. However, most research focuses on overall performance of the network without delving into its internal representations. In this work, we examine how the internal representations are influenced by data quality, leveraging this analysis to gain deeper insights into robustness and memorization. Specifically, we investigate two noisy data settings: label noise and corrupted inputs.

For DNNs trained on a noisy dataset (with either noisy labels or noisy inputs), we evaluate their internal representation learning abilities on both noisy and clean data, specifically using $\mathcal{NC}_1^{\text{clean},l}$ and $\mathcal{NC}_1^{\text{noise},l}$ that represent the $l$-th layer $\mathcal{NC}_1$ computed on the clean and noisy datasets, respectively. We now introduce the notion of *memorization ratio* based on $\mathcal{NC}_1^{\text{clean},l}$ and $\mathcal{NC}_1^{\text{noise},l}$.

**Definition 1 (Memorization ratio)** *For a DNN trained on noisy data, we call the ratio* $\Delta_{\mathcal{NC}_1^l} = \frac{\mathcal{NC}_1^{clean,l}}{\mathcal{NC}_1^{noise,l}}$ *as the memorization ratio of the network at the l-th layer.*

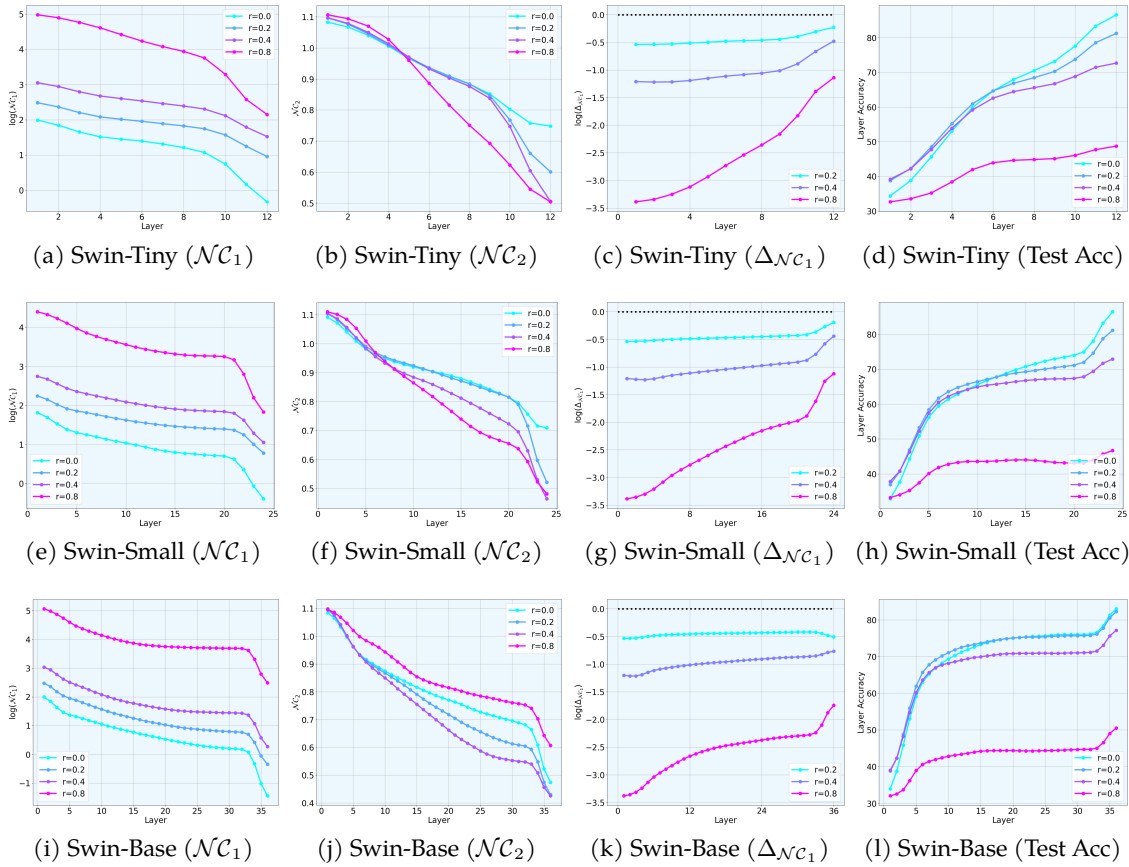

| (a) Swin-Tiny ($\mathcal{NC}_1$) | (b) Swin-Tiny ($\mathcal{NC}_2$) | (c) Swin-Tiny ($\Delta_{\mathcal{NC}_1}$) | (d) Swin-Tiny (Test Acc) |
|---|---|---|---|
| (e) Swin-Small ($\mathcal{NC}_1$) | (f) Swin-Small ($\mathcal{NC}_2$) | (g) Swin-Small ($\Delta_{\mathcal{NC}_1}$) | (h) Swin-Small (Test Acc) |
| (i) Swin-Base ($\mathcal{NC}_1$) | (j) Swin-Base ($\mathcal{NC}_2$) | (k) Swin-Base ($\Delta_{\mathcal{NC}_1}$) | (l) Swin-Base (Test Acc) |

Figure 4: **The intermediate $\mathcal{NC}$, memorization ratio and linear-probing accuracy of Swin-Transformer models on random labeled CIFAR-10 dataset.**

Intuitively speaking, when the feature mapping overfits the noisy dataset, $\mathcal{NC}_1^{\text{noise},l}$ becomes small while $\mathcal{NC}_1^{\text{clean},l}$ remains large, resulting in a high memorization ratio $\Delta_{\mathcal{NC}_1^l}$. Conversely, if the feature mapping encodes meaningful features, the memorization ratio $\Delta_{\mathcal{NC}_1^l}$ will be small. It is important to note that the memorization ratio is unsuitable for comparisons between models trained with different noise levels, as $\mathcal{NC}_1^{\text{noise},l}$ is computed on varying noisy training data. Building on this notion, we now introduce the concept of memorization layers as follows.

**Definition 2 (Memorization layers)** *For a DNN trained on noisy data, we define the memorization layers as $\{l : \Delta_{\mathcal{NC}_1^l} > 1\}$, which often occur consecutively and primarily in the final few layers.*

**Label noise:** We utilize a synthetically generated, randomly labeled CIFAR-10 dataset, and the real-world, human-annotated CIFAR-10N dataset [52] to study the effects of label noise. Our objective is not to propose improved training methods or architectures for handling label noise but to gain a deeper understanding of how label noise impacts representation learning under standard training.

**1. No, deeper layers are also adept at enhancing neural collapse and more prone to memorization.**
Figure 3 and Figure 4 visualize intermediate neural collapse across varying noise levels using a ResNet model and Swin-Transformer models on the randomly labeled CIFAR-10 dataset. Additional results for the human annotated CIFAR-10N dataset (Figure 8 and Figure 9) are provided in the Appendix. From the first two columns of figures, we observe consistent patterns: improvements in within-class variability and between-class separation across different noise levels, similar to trends observed with clean labels. However, as the proportion of noisy data increases, the overall within-class compression curves shift upward, indicating the increasing challenge for the model to collapse semantically different features together. While previous work has demonstrated that features in the

penultimate layer exhibit neural collapse even with purely random-labeled data [16], our findings reveal that the model's layers internally recognize noise, and intermediate neural collapse emerges as an effective indicator of the noise level.

Figure 3 (third column) depicts the layer-wise memorization ratio, $\Delta_{\mathcal{NC}_1^l}$, computed between clean and noisy data. The results demonstrate a progressive increase in the memorization ratio across layers, irrespective of network size, suggesting that memorization predominantly occurs in the final layers. Analyzing the impact of noise levels on linear probing accuracy using clean test data reveals that the performance gap between models pre-trained with varying degrees of label noise is small in the initial layers but widens in the deeper layers. As noise levels increase, more deeper layers exhibit a declining trend in performance on clean data. This suggests that the initial layers primarily learn noise-agnostic, general features, while the later layers focus on task-specific features, rendering them more vulnerable to noise.

**2&3. No, deeper Swin-Transformer models suffer less from memorization, while deeper ResNet models are more susceptible on noisy label.** When comparing different architectures (Figure 3 and Figure 4), we identify two key differences between ResNets and Swin-Transformers in terms of their memorization ratios. First, in Swin-Transformers, the memorization ratio consistently remains below 1 across all layers, reflecting their ability to focus on learning meaningful data structures rather than memorizing incorrect labels. In contrast, ResNets exhibit memorization ratios exceeding 1 in the final layers, indicating a propensity to overfit noisy labels at deeper layers. Second, at a specific noise level, increasing the model depth leads to higher memorization ratios in ResNets, suggesting an amplified tendency to memorize noisy labels. Conversely, deeper Swin-Transformer models demonstrate decreasing memorization ratios, indicating enhanced robustness to label noise as depth increases. As a result, increasing model depth improves the performance of Swin-Transformer models but degrades the performance of ResNet models on clean data.

This contrast likely stems from the strong design priors in ResNets, which, while effective at extracting task-relevant features, make deeper models more prone to overfitting on noisy data. In comparison, Transformer-based models such as Swin-Transformers inherently leverage intrinsic data correlations to learn robust and meaningful representations, reducing the susceptibility of deeper models to label noise. These findings align with prior research [53], which demonstrates that Transformer-based models excel at capturing complex hierarchical data correlations, and further corroborate their superior robustness to noisy labels through improved feature representation [54].

**Corrupted input data:** To investigate the impact of corrupted input data, we utilize the CIFAR-10C dataset [55], which includes various common perturbations. We visualize intermediate neural collapse across different noise levels for Gaussian-type input perturbations in Figure 10, Figure 11. Our results reveal that Similar to the noisy label case, **1. deeper layers are also adept at enhancing neural collapse and more prone to memorization.** However, unlike the noisy label, 2. **deeper ResNet models suffer less from memorization, while deeper Swin-Transformer models are more susceptible on corrupted input.** Due to the page limitation, please refer Appendix A.3 in appendix for more details and discussion.

## 5. Conclusion

In this work, we extend the previous study of progressive neural collapse to larger models and more complex datasets, offering a comprehensive understanding of its role in generalization and robustness. By analyzing intermediate representations under clean and noisy data conditions, our findings reveal three key insights: First, deeper layers excel in enhancing neural collapse, playing a critical role in generalization improvement but are also more prone to memorization. Second, in deeper networks, middle layers show diminished neural collapse enhancement, leading to redundancy. Last, compared to convolutional models, Transformer models demonstrate superior neural collapse enhancement and robustness on more complex datasets. These findings enhance our understanding of the inner mechanisms of deep neural networks and provide new perspective for designing more efficient and robust deep learning models.

## Acknowledgements

We gratefully acknowledge support from NSF grants CCF-2240708, IIS-2312840, and IIS-2402952, as well as the ORAU Ralph E. Powe Junior Faculty Enhancement Award. We also extend our sincere appreciation to Chong You (Google Research), Qing Qu (University of Michigan), and Jeremias Sulam (Johns Hopkins University) for their insightful and valuable discussions.

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

# A. Appendix

In the appendix, we provide additional details and experiments to complement the main paper. In Appendix A.1, we begin by outlining the experimental setup and details necessary for reproduction. Then, in Appendix A.2, we present the results of supplementary experiments. Finally, we present the results and discuss the impacts of corrupted input in Appendix A.3.

## A.1. More experiments details.

**Details of Figure 1 and Figure 5.** For the CIFAR-10 and CIFAR-100 datasets, we follow standard practices by normalizing the images channel-wise using their mean and standard deviation. A standard data augmentation strategy is employed, including random cropping to $32 \times 32$ with a padding of 4, followed by random horizontal flipping with a probability of 0.5. During pre-training, the network is trained for 200 epochs with a batch size of 128 using the SGD optimizer with a momentum of 0.9 and weight decay of $0.05$. The initial learning rate is set to 0.1 and is reduced by a factor of 10 at epochs 100 and 150. Similar to the penultimate layers, linear probing is performed on intermediate features from intermediate layers, followed by an adaptive average pooling operation that reduces the spatial dimensions to $1 \times 1$. For linear probing, the backbone model is frozen, and only an additional linear classifier is trained across residual blocks. The linear classifier on the intermediate features is trained for 10 epochs using the same SGD parameters as in the pre-training stage, except the learning rate is reduced by a factor of 5 at epochs 5 and 7. For ImageNet, we utilize the pretrained weights provided by the mmpretrain library [56]. Similar to the CIFAR datasets, linear probing is conducted by freezing the backbone model and training only an additional linear classifier across residual blocks. However, the data augmentation strategy follows the practices described in [56], as used during pretraining on ImageNet. Additionally, the $\mathcal{NC}$ is calculated on the test dataset.

**Details of Figure 2 and Figure 6.** For the CIFAR-10 and CIFAR-100 datasets, we adhere to standard preprocessing practices by normalizing the images channel-wise using their respective means and standard deviations. Data augmentation includes random cropping to a size of $32 \times 32$ with a padding of 4, followed by random horizontal flipping with a probability of 0.5. To analyze the impact of depth, the Base Swin-Transformer is modified by increasing the number of blocks in each stage from [2, 2, 18, 2] to [2, 2, 30, 2] and reducing the number of attention heads per stage from [4, 8, 16, 32] to [3, 6, 12, 24]. Similarly, for the Large Swin-Transformer, the number of blocks is increased from [2, 2, 18, 2] to [2, 2, 42, 2], and the number of attention heads is reduced from [6, 12, 24, 48] to [3, 6, 12, 24]. During pre-training, the network is trained for 200 epochs with a batch size of 128, using the AdamW optimizer configured with $\beta_1 = 0.9$, $\beta_2 = 0.999$, and and a weight decay of 0.05 applied to all parameters except the class token and positional embeddings. The initial learning rate is set to $10^{-4}$ and and follows a linear warm-up schedule, increasing to $10^{-3}$ bby epoch 20. Subsequently, a cosine decay schedule reduces the learning rate gradually to $10^{-5}$ by the end of training. Linear probing is performed on intermediate features extracted from intermediate layers, followed by an adaptive average pooling operation to reduce the spatial dimensions to $1 \times 1$. For linear probing, the backbone model is frozen, and only an additional linear classifier is trained across residual blocks. The linear classifier for the intermediate features is trained for 10 epochs using the same SGD parameters as in the linear-probing stage of ResNet, with the learning rate reduced by a factor of 5 at epochs 5 and 7. For ImageNet, we use the pretrained weights of the Tiny and Small Swin-Transformers provided by the mmpretrain library [56] and follow the original configurations to train the Base Swin-Transformer. The data augmentation strategy aligns with the practices described in [56] and it used in ResNet linear-probing stage.

**Details of Figure 3 and Figure 4.** In the main body of this paper, we present results on the synthesized randomly labeled CIFAR-10 dataset. The training configuration remains identical to that used for ResNet and Swin-Transformer models on the clean CIFAR-10 dataset, except that a certain percentage of labels are randomly shuffled and fixed throughout the training process.

**Details of Figure 8 and Figure 9.** In the appendix, we present results obtained on the CIFAR-10N dataset. Unlike synthetic noise (e.g., random noise), CIFAR-10N contains human-annotated label

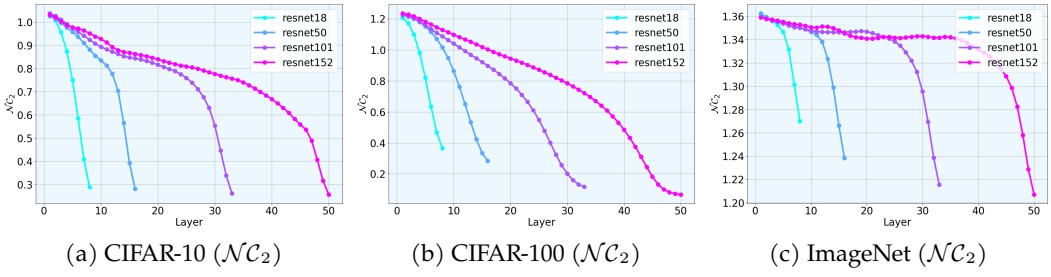

(a) CIFAR-10 ($\mathcal{NC}_2$)      (b) CIFAR-100 ($\mathcal{NC}_2$)      (c) ImageNet ($\mathcal{NC}_2$)

Figure 5: **The intermediate $\mathcal{NC}_2$ of ResNet models on different datasets.**

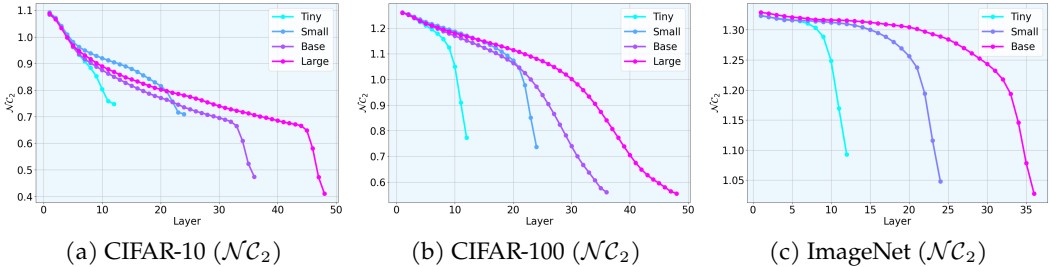

(a) CIFAR-10 ($\mathcal{NC}_2$)      (b) CIFAR-100 ($\mathcal{NC}_2$)      (c) ImageNet ($\mathcal{NC}_2$)

Figure 6: **The intermediate $\mathcal{NC}_2$ of different Swin-Transformer models on different dataset.**

noise, which reflects practical annotation errors. The noisy labels were collected by aggregating annotations from multiple human annotators, resulting in different subsets that allow researchers to evaluate the impact of varying noise levels. For this study, we adopt three noise types: Aggregate, Random 1, and Worst, with noise rates of $9.03\%$, $17.23\%$, and $40.21\%$, respectively. Aggregate (aggre) combines annotations from multiple annotators to minimize random errors and serves as a relatively realistic noise representation. Random 1 (randn1) represents noisy labels assigned entirely at random, simulating unstructured noise where human annotators randomly guessed labels. Worst refers to the subset of labels with the highest error rates, representing the most challenging and biased noise scenario. The training configurations used for ResNet and Swin-Transformer models on the noisy CIFAR-10N dataset are identical to those applied to the clean CIFAR-10 dataset.

**Details of Figure 10 Figure 11 and Figure 12.** In the appendix, we present results obtained on the CIFAR-10C dataset, a corruption-augmented version of the original CIFAR-10 dataset designed to benchmark the robustness of machine learning models against various types of common corruptions. CIFAR-10C includes 15 types of corruptions commonly encountered in real-world scenarios, each with 5 severity levels, ranging from mild to extreme. This setup enables a gradual assessment of model robustness under increasing levels of difficulty. In this study, we focus on two specific corruption types: Gaussian noise and Speckle noise, to analyze the impact of corrupted inputs on intermediate representations. We consider three severity levels, level 1, level 3, and level 5, and for continuity, reassign these as level 1, level 2, and level 3 in our analysis. The training configurations used for ResNet and Swin-Transformer models on the noisy CIFAR-10C dataset are identical to those applied to the clean CIFAR-10 dataset.

## A.2. More experiments results.

**Additional Results of $\mathcal{NC}_2$ on the Clean Dataset.** As a supplement to the intermediate $\mathcal{NC}_1$ results presented in Figure 1 and Figure 2 for the clean dataset, we additionally provide the intermediate $\mathcal{NC}_2$ results. These are shown for ResNet models in Figure 5 and for Swin-Transformer models in Figure 6. From the figures, we consistently observe trends similar to those in $\mathcal{NC}_1$: 1.Deeper layers exhibit a more pronounced enhancement in neural collapse compared to shallower layers. 2. As model depth increases, the middle layers show a slower trend in neural collapse enhancement. While the saturated plateau observed in deeper layers disappears on the CIFAR dataset, it re-emerges on the ImageNet dataset. This is because, in datasets with a large number of classes, the

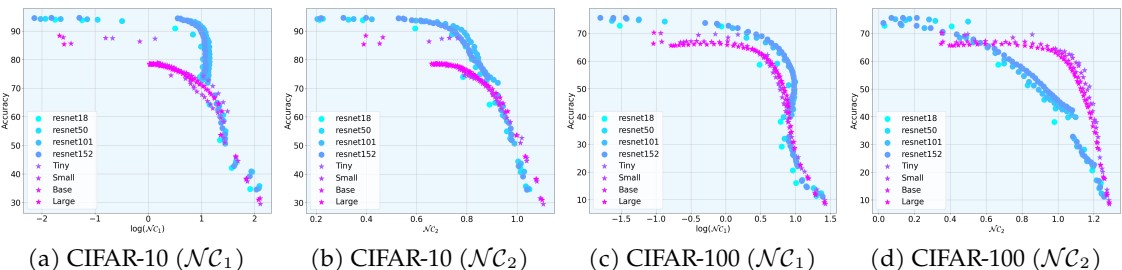

(a) CIFAR-10 ($\mathcal{NC}_1$)    (b) CIFAR-10 ($\mathcal{NC}_2$)    (c) CIFAR-100 ($\mathcal{NC}_1$)    (d) CIFAR-100 ($\mathcal{NC}_2$)

Figure 7: **The evolution of intermediate $\mathcal{NC}$ for different Swin-Transformer based models.**

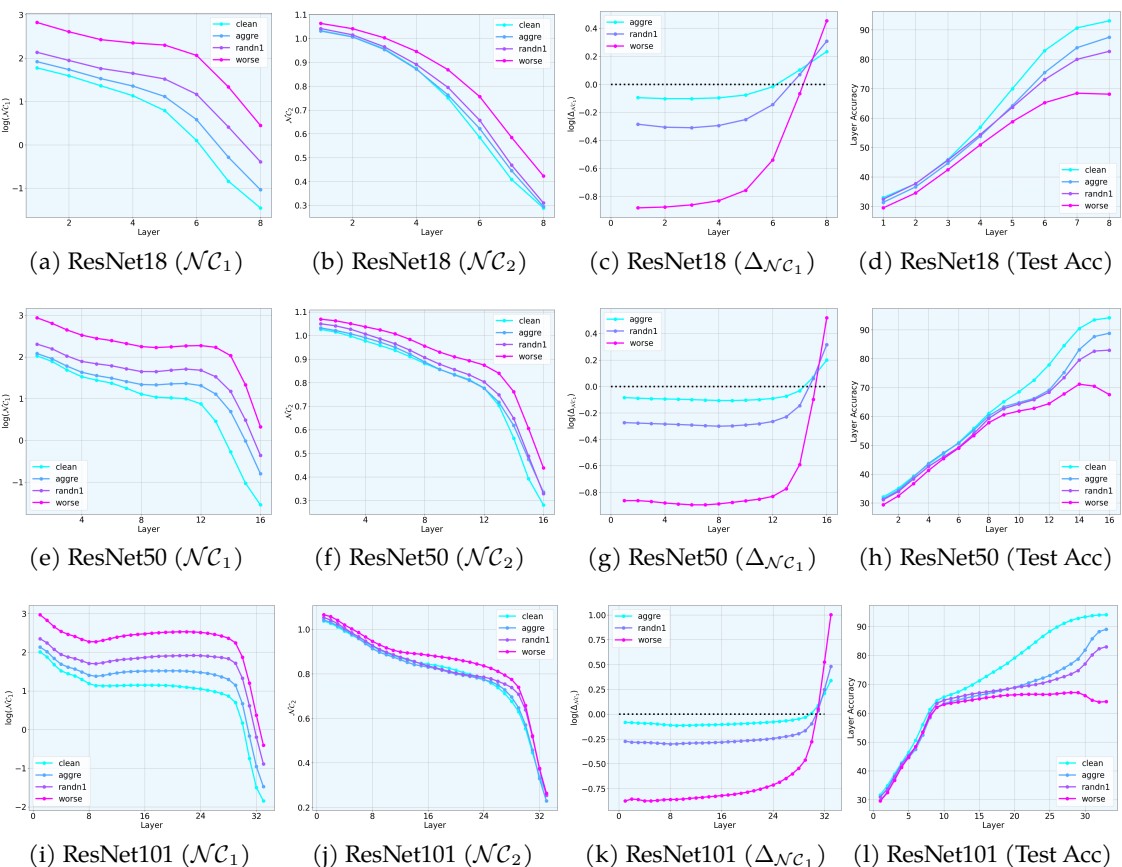

(a) ResNet18 ($\mathcal{NC}_1$)   (b) ResNet18 ($\mathcal{NC}_2$)   (c) ResNet18 ($\Delta_{\mathcal{NC}_1}$)   (d) ResNet18 (Test Acc)

(e) ResNet50 ($\mathcal{NC}_1$)   (f) ResNet50 ($\mathcal{NC}_2$)   (g) ResNet50 ($\Delta_{\mathcal{NC}_1}$)   (h) ResNet50 (Test Acc)

(i) ResNet101 ($\mathcal{NC}_1$)   (j) ResNet101 ($\mathcal{NC}_2$)   (k) ResNet101 ($\Delta_{\mathcal{NC}_1}$)   (l) ResNet101 (Test Acc)

Figure 8: **The intermediate $\mathcal{NC}$, memorization ratio and linear-probing accuracy of ResNet models on CIFAR-10N dataset.** The graphs depict the layer-wise progression of within-class compression (first column), between-class separation (second column) using noisy label, memorization ratio $\Delta_{\mathcal{NC}_1}$ (third column) and layerwise linear-probing accuracy (last column). The percentage of noisy labels increases in the order: clean, aggre, randn1, worse.

class-mean features are more uniformly distributed, and slight perturbations in these features may not be reflected as clearly in the $\mathcal{NC}_2$ metric. This explains why $\mathcal{NC}_1$ serves as a more sensitive indicator of generalization performance than $\mathcal{NC}_2$. 3. Transformer models outperform convolutional models in enhancing neural collapse on larger datasets, indicating a positive correlation between neural collapse and generalization performance.

**Additional Results on Noisy Labels in the CIFAR-10N Dataset.** In Figure 8 and Figure 9, we present additional results on the CIFAR-10N dataset, which captures practical human annotation errors. Notably, the error rates vary across classes, reflecting an imbalanced distribution of labeling errors. Despite this imbalance, we consistently observe trends similar to those seen in the synthetic

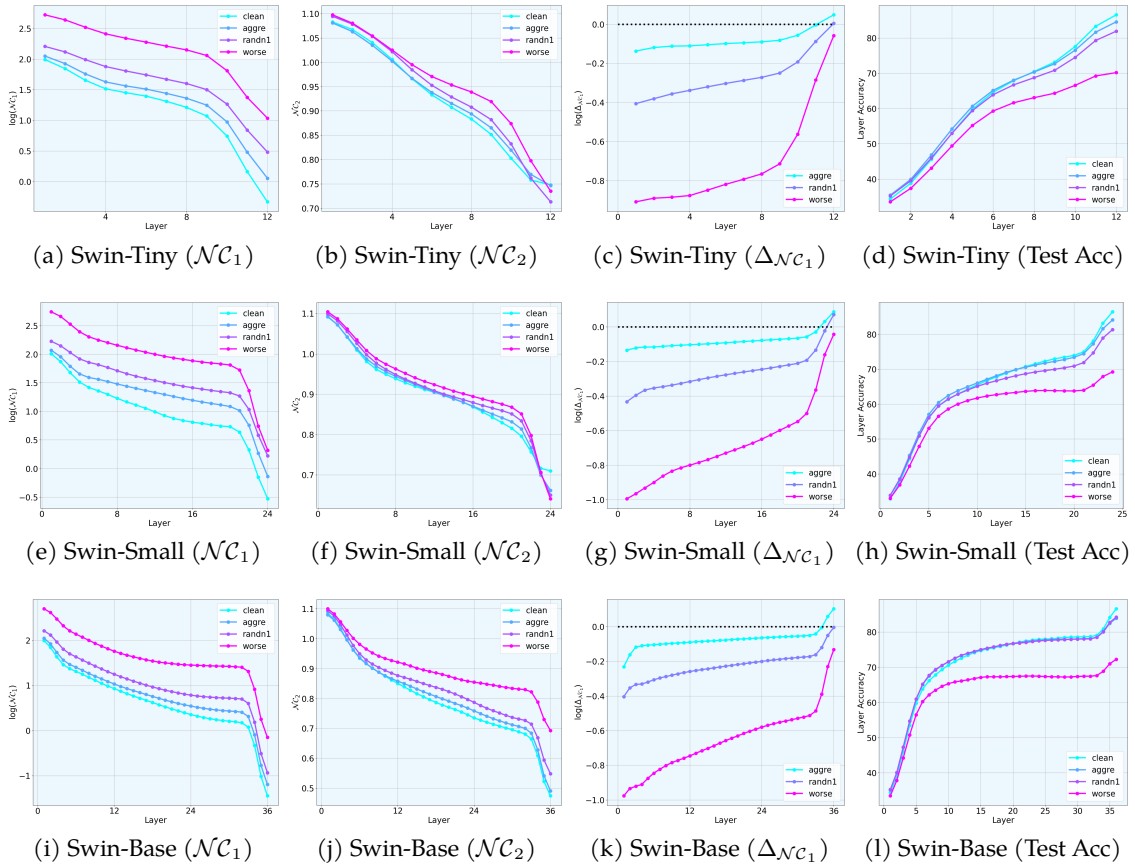

Figure 9: **The intermediate $\mathcal{NC}$, memorization ratio and linear-probing accuracy of Swin-Transformer models on CIFAR-10N dataset.** The percentage of noisy labels increases in the order: clean, aggre, randn1, worse.

randomly shuffled CIFAR-10 dataset: 1. Deeper layers also excel at enhancing neural collapse but more prone to memorization. 2. Deeper Transformer models exhibit reduced memorization compared to convolutional models, whereas deeper convolutional models show an increasing tendency toward memorization. These findings suggest that the observed impact of noisy labels on intermediate representations is a universal phenomenon.

## A.3. Are All Layers Created Equal under Corrupted Input?

In this section, we investigate the impact of corrupted input on intermediate features and its relationship with memorization and robustness, addressing the three key questions outlined in Section 4:

1. Are layers at different depths created equal?
2. Are layers in models of varying depths created equal?
3. Are layers from different architectures created equal?

**1. No, deeper layers are adept at enhancing neural collapse but are also more prone to memorization.** Figure 10 and Figure 11 illustrate the progression of intermediate neural collapse across varying noise levels using ResNet and Swin-Transformer models on the Gaussian-corrupted CIFAR-10C dataset. From the first two columns of these figures, we observe consistent patterns: enhancements in within-class variability and between-class separation across different noise levels, similar to the trends observed with clean labels. However, as the proportion of noisy data increases, the within-class compression curves shift upward, highlighting the growing difficulty for the model to collapse semantically distinct features effectively. These findings suggest that the model's layers adapt inter-

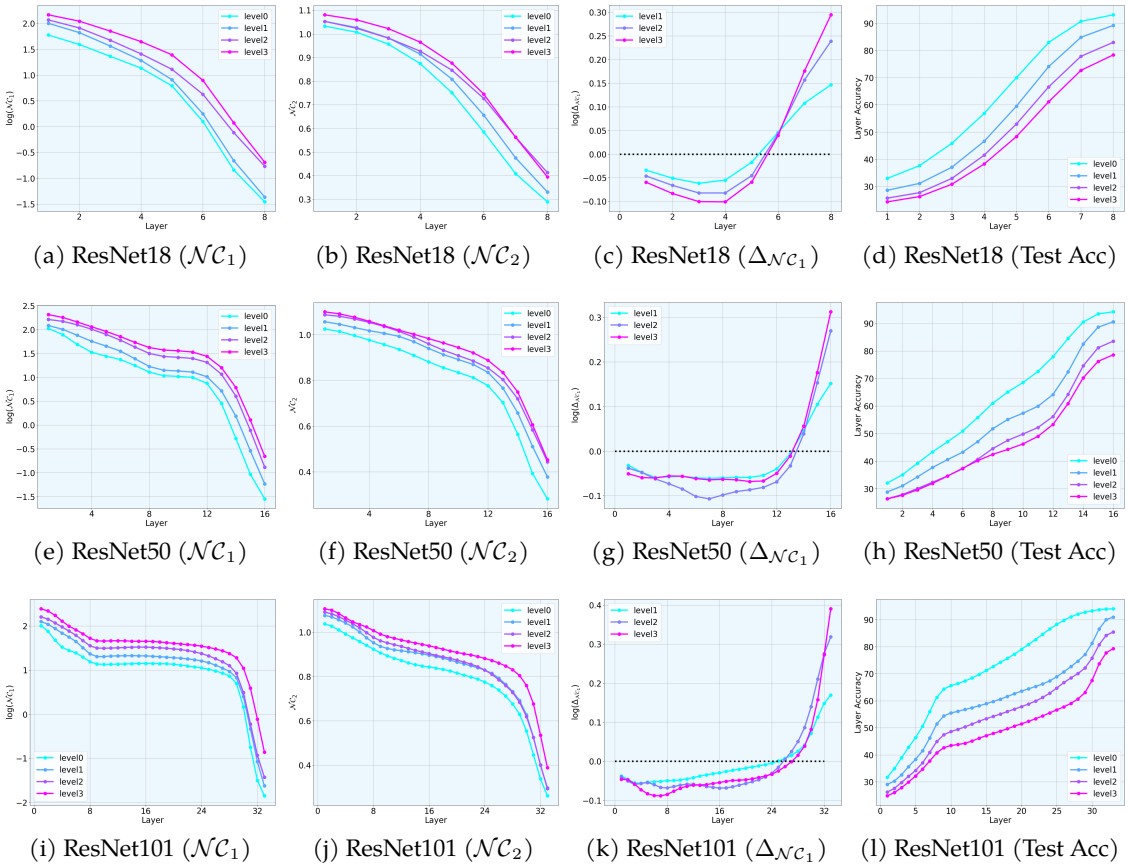

Figure 10: **Intermediate $\mathcal{NC}$, Memorization Ratio, and Linear-Probing Accuracy of ResNet Models on the CIFAR-10C (Gaussian) Dataset.** The graphs illustrate the layer-wise progression of within-class compression (first column), between-class separation (second column), memorization ratio $\Delta_{\mathcal{CDNV}}$ (third column), and layer-wise linear-probing accuracy (last column) for different ResNet architectures on the CIFAR-10C [55] dataset with Gaussian noise. The degree of corruption increases in the order: level0 → level3, where level0 represents the clean data.

nally to recognize corrupted input, with intermediate neural collapse serving as a reliable indicator of noise levels. This observation aligns with the idea that deep neural networks inherently possess image priors, enabling them to denoise various types of corrupted input [57, 58].

Figure 10 and Figure 11 (third column) reveal a decreasing trend in memorization across the initial layers, followed by a progressive increase in the memorization ratio in the subsequent layers, irrespective of network size. This indicates that the early layers effectively filter out certain noise, while the deeper layers are more prone to memorizing the remaining noisy features. Further analysis of the impact of noise levels on linear probing accuracy using clean test data shows that the performance gap between models pre-trained with different degrees of label noise is minimal in the initial layers but becomes increasingly pronounced in the deeper layers. As noise levels rise, more deeper layers exhibit a decline in performance on clean data. These findings also support that the initial layers primarily learn noise-agnostic, general features, whereas the deeper layers focus on task-specific features, making them more susceptible to noise.

When comparing different architectures (Figure 3 and Figure 4), we identify two key differences between ResNets and Swin-Transformers in terms of their memorization ratios. First, while Swin-Transformers initially demonstrate a smaller memorization ratio compared to ResNet models in the early layers, this ratio increases more rapidly as the layers deepen. This suggests that corrupted input has a more detrimental effect on the quality of features learned in the initial layers of Swin-Transformers. Consequently, the deeper layers of Swin-Transformers tend to memorize these low-

quality features to fit the task, resulting in a larger discrepancy at the deeper layers. In contrast, ResNets show a larger initial memorization ratio but a slower rate of increase as layers deepen. This behavior may be attributed to the convolutional layers' inherent design priors, which enable them to filter out corrupted noise more effectively. As a result, the remaining higher-quality features, which are closer to clean features, are passed to subsequent layers for processing, leading the memorization ratio closer to 0. Second, at a specific noise level, increasing model depth in Swin-Transformer models results in a more pronounced increase in memorization ratios, highlighting an amplified tendency to overfit these low-quality features. Conversely, deeper ResNet models exhibit only negligible increases in memorization ratios, indicating greater robustness to corrupted input with increased depth. Consequently, while increasing model depth degrades the performance of Swin-Transformer models on clean data, it enhances the performance of ResNet models slightly.

This contrast also highlights the differences in learning mechanisms between convolutional models and Swin-Transformer models. It supports the widely held belief that Transformer-based models, such as Swin-Transformers, inherently leverage intrinsic data correlations to learn meaningful representations. However, when the input is corrupted, uncovering meaningful data structures becomes significantly more challenging, leading to substantial impairment of the learned representations. In comparison, the strong convolutional priors in ResNets effectively smooth the input data, reducing their susceptibility to corrupted input and preserving the quality of the learned features.

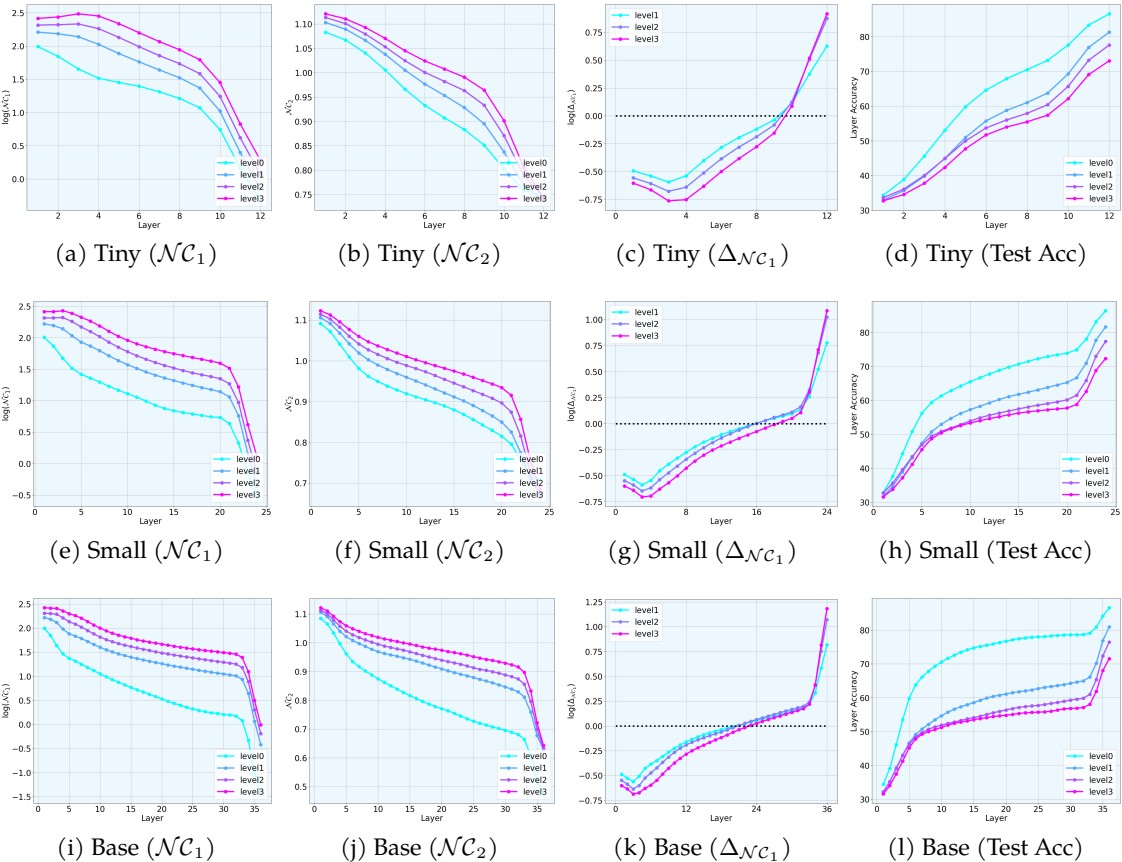

Figure 11: **Intermediate $\mathcal{NC}$, Memorization Ratio, and Linear-Probing Accuracy of Swin-Transformer Models on the CIFAR-10C (Gaussian) Dataset.** The graphs illustrate the layer-wise progression of within-class compression (first column), between-class separation (second column), memorization ratio $\Delta_{\mathcal{CD}\mathcal{NV}}$ (third column), and layer-wise linear-probing accuracy (last column) for different Swin-Transformer architectures on the CIFAR-10C [55] dataset with Gaussian noise. The degree of corruption increases in the order: level0 → level3, where level0 denotes clean input.

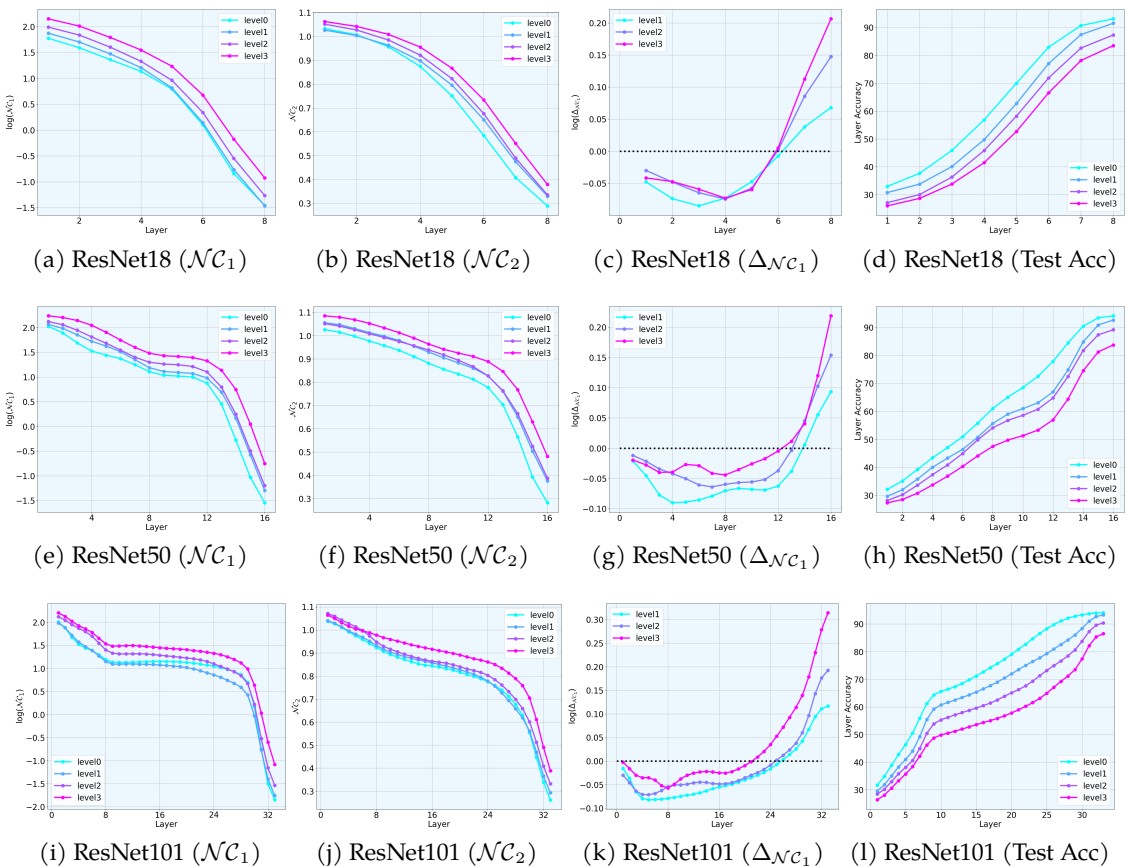

Figure 12: **Intermediate $\mathcal{NC}$, Memorization Ratio, and Linear-Probing Accuracy of ResNet Models on the CIFAR-10C (Speckle) Dataset.** The graphs illustrate the layer-wise progression of within-class compression (first column), between-class separation (second column), memorization ratio $\Delta_{\mathcal{CDNV}}$ (third column), and layer-wise linear-probing accuracy (last column) for different ResNet architectures on the CIFAR-10C [55] dataset with Speckle noise. The degree of corruption increases in the order: level0 → level3, where level0 represents the clean data.

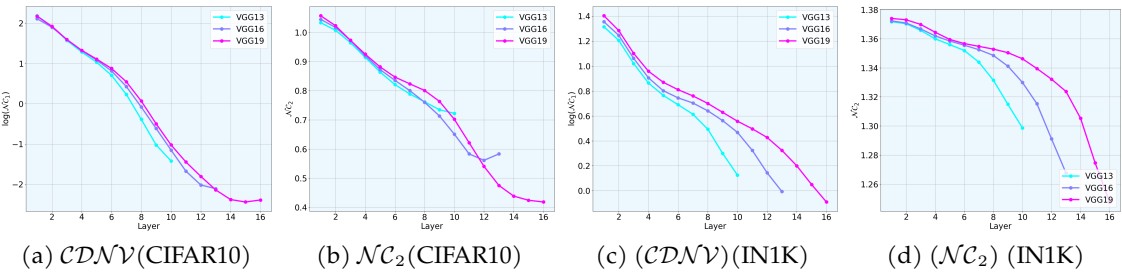

Figure 13: **Intermediate $\mathcal{CDNV}$, and $\mathcal{NC}_2$ of VGG Models on the CIFAR10 and ImageNet-1k(In1k) Dataset.**

# B. Additional Experiments on VGG

Additionally, we conducted further experiments on intermediate neural collapse using VGG models, including VGG13, VGG16, and VGG19, as shown in Figure 13. The results reveal an almost geometric decay rate of CDNV on both the CIFAR10 and ImageNet datasets. Notably, the decay rate decreases as model depth increases, consistent with previous observations in [10] and aligning with the results for shallower models, such as ResNet18, shown in Figure 1. These findings further support our claim that when model complexity is lower than dataset complexity, CDNV exhibits a

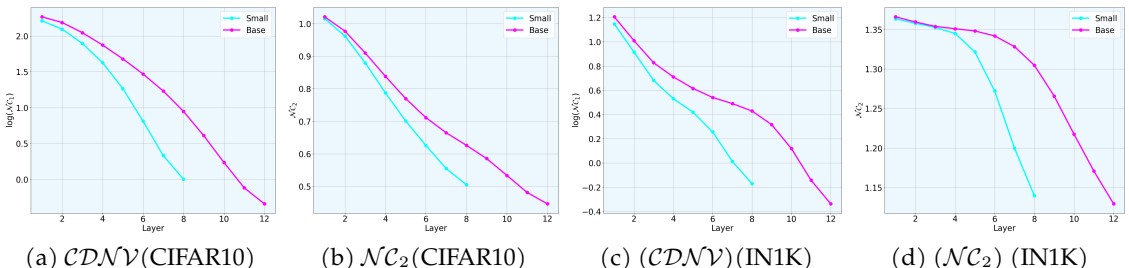

(a) $\mathcal{CDNV}$(CIFAR10)  (b) $\mathcal{NC}_2$(CIFAR10)  (c) ($\mathcal{CDNV}$)(IN1K)  (d) ($\mathcal{NC}_2$) (IN1K)

Figure 14: **Intermediate $\mathcal{CDNV}$, and $\mathcal{NC}_2$ of Vision Transformer Models on the CIFAR10 and ImageNet-1k(In1k) Dataset.**

geometric decay rate of within-class compression across layers. However, due to the lack of residual connections, it is challenging to study the evolution of neural collapse across layers in deeper models.

# C. Additional Experiments on Vision Transformer

In Figure 14, we present the intermediate neural collapse results for Vision-Transformer (ViT) models on both the CIFAR10 and ImageNet datasets. Notably, without the design prior, the last-layer training accuracy of ViT-small and ViT-base fails to reach the neural collapse regime on the small CIFAR10 dataset, where the training accuracy approaches 100%. This indicates that the model complexity of ViT-small and ViT-base is insufficient to fully capture the complexity of the dataset. As a result, both the $\mathcal{CDNV}$ and $\mathcal{NC}_2$ metrics exhibit a continuous, geometric-rate decreasing trend. Furthermore, the weaker intermediate compression and separation observed in ViT models on the CIFAR10 dataset suggest that these architectures are less efficient at extracting and organizing meaningful features from smaller datasets. However, as the dataset size increases to ImageNet, ViT models demonstrate improved intermediate compression and separation, with diminished neural collapse at the middle layers. This indicates that ViT architectures are more effective at extracting and organizing meaningful features from larger datasets such as ImageNet. These observations align with findings reported in [59] and our claims in Section 4.1.

