# OpenReview forum: "Are all layers created equal: A neural collapse perspective"
_CPAL.cc/2025/Proceedings_Track — CPAL 2025 (Proceedings Track) Poster_

### Official Review · Reviewer_czKY · 2025-01-03

**Rating:** 6
**Confidence:** 3

**Review:**

### Summary

This paper extends the previous study of progressive neural collapse in larger models and complex datasets, including both clean and noisy data. The findings reveal three key insights: (1) Layer inequality. (2)  Depth dependent behavior. (3) Architectural differences.

### 1. Quality
The argument is clear, but there is insufficient evidence to support it.

### 2. Clarity
The writing effectively communicates the intended message and is easy to understand.

### 3. Originality
The paper extends the previous work by increasing the model size and dataset complexity and claims to reveal findings not identified in prior studies. However, the lack of thorough experimentation raises doubts about the validity of these claims. Therefore, the novelty of the work seems limited.

### 4. Significance
Gaining insight into how features develop across layers from the neural collapse perspective is important for revealing the inner mechanisms of deep neural networks.

### Conclusion

- **Strengths**
  - The paper makes an effort to expand the scope of experiments compared to prior work.
  - The writing is clear.

- **Weaknesses**
  - The exclusion of VGG from the experiments, due to its relatively small architecture, is questionable. The paper claims VGG is smaller than ResNet, but this may not be accurate. It is unclear why VGG was excluded from the experiments.
  - To support the third claim, a wider range of Transformer models should have been tested, at least including ViT.
  - The experimental analysis is somewhat lacking. There seems to be a leap in the arguments made from the results, and more thorough analysis and theoretical validation are needed to support these claims.

---

### Official Review · Reviewer_exyr · 2025-01-07

**Rating:** 7
**Confidence:** 3

**Review:**

**Summary**: This paper studied how features evolved through layers in deep neural network via a neural collapse perspective. The authors found the following: (1) deeper layers significantly enhance neural collapse (2) middle layers contribute minimally to neural collapse and (3) the behavior of layers are different across architectures (resnet vs transformer in this work).

**Clarity and originality**: This work is clearly presented and is original.

**Pros**: I found the results on how different layers contributing to neural collapse (Figure 1 and 2) very intriguing. The results are clearly presented. In particular, Figure 1 showed that in deep ResNet, the middle layers have minimal contributions to neural collapse and the final few layers contributes significantly. Figure 2 showed that transformers have different behaviors from ResNet. I also like the authors' experiment on doing linear probing on each layer. Overall, I found the results presented in this work interesting and are good contributions to the field.

**Cons**: I didn't find any obvious flaws of this work.

**Disclaimer**: I don't work on neural collapse and thus my judgement is based on the common knowledge of neural collapse. Based on my current knowledge, I think this work made some interesting contributions.

---

### Official Review · Reviewer_rHvu · 2025-01-10
**Reviews**

**Rating:** 7
**Confidence:** 3

**Review:**

This paper investigates the phenomenon of Neural Collapse (NC) across the layers of deep neural networks. It extends prior research by studying NC in larger architectures and complex datasets, including noisy data settings.

**Strengths:**

1. The paper explores how neural collapse (NC) evolves across layers in deep networks,and it is interesting.

2. Comprehensive empirical analysis using diverse datasets and architectures (ResNet, Swin-Transformer).

3. This paper provides Key insights all different layer, include deeper layers enhancing NC, middle-layer redundancy, and Transformers being more robust to memorization.


**Questions:**

1. The reasons why deeper layers enhance NC more than middle layers remain unclear. Can you bring some explanations?

2. Lack of statistical significance reporting (e.g., error bars, confidence intervals).

3. How exactly does the Class-Distance Normalized Variance (CDNV) relate to neural collapse compared to the original NC1 and NC2 metrics?

---

### Meta-Review · Area_Chair_J924 · 2025-01-29

**Recommendation:** Accept (Poster)
**Confidence:** 5

**Metareview:**

The paper provides a detailed investigation into the evolution of neural collapse across layers in deep networks, extending prior work to larger architectures and complex datasets. The study offers key insights into layer inequality, depth-dependent behavior, and architectural differences, with findings that deepen our understanding of generalization, memorization, and robustness. While some concerns were raised regarding statistical significance and broader model comparisons, the authors provided satisfactory rebuttals and additional experiments addressing these issues. Given the strong empirical contributions and the overall positive reception, I recommend acceptance.

As an additional note, although the paper addresses the literature on intermediate neural collapse and proposed various novel observations, many of its observations are similar to findings that have appeared in previous contributions (e.g., NC1 improving as we move up in the network, the behavior of NC as a reaction to noisy labels, etc.). The section on Intermediate Neural Collapse in the related work could better acknowledge this prior research. It would be particularly valuable for the authors to engage more meaningfully with the existing literature by explicitly comparing their results with pre-existing observations.

---

### Decision · Program_Chairs · 2025-02-11

Accept (Poster)